# Full-Length Transcriptome Sequencing Reveals the Impact of Cold Stress on Alternative Splicing in Quinoa

**DOI:** 10.3390/ijms23105724

**Published:** 2022-05-20

**Authors:** Ling Zheng, Yiwu Zhao, Yifeng Gan, Hao Li, Shiqi Luo, Xiang Liu, Yuanyuan Li, Qun Shao, Hui Zhang, Yanxiu Zhao, Changle Ma

**Affiliations:** 1Shandong Provincial Key Laboratory of Plant Stress, Life Science College, Shandong Normal University, Jinan 250014, China; zhengling0000@sina.com (L.Z.); zywkaka@163.com (Y.Z.); gyf199506@126.com (Y.G.); sdsqflh@163.com (H.L.); luoshiqi0010@163.com (S.L.); liuxiang182674@163.com (X.L.); shaoqun2000@hotmail.com (Q.S.); laohanzhang@hotmail.com (H.Z.); 2CAS Center for Excellence in Molecular Plant Sciences, Institute of Plant Physiology and Ecology, Chinese Academy of Sciences, Shanghai 200032, China; yyli@cemps.ac.cn

**Keywords:** quinoa, full-length transcriptomes, cold stress, ROS homeostasis

## Abstract

Quinoa is a cold-resistant and nutrient-rich crop. To decipher the cold stress response of quinoa, the full-length transcriptomes of the cold-resistant quinoa variety CRQ64 and the cold-sensitive quinoa variety CSQ5 were compared. We identified 55,389 novel isoforms and 6432 novel genes in these transcriptomes. Under cold stress, CRQ64 had more differentially expressed genes (DEGs) and differentially alternative splicing events compared to non-stress conditions than CSQ5. DEGs that were specifically present only in CRQ64 were significantly enriched in processes which contribute to osmoregulation and ROS homeostasis in plants, such as sucrose metabolism and phenylpropanoid biosynthesis. More genes with differential alternative splicing under cold stress were enriched in peroxidase functions in CRQ64. In total, 5988 transcription factors and 2956 long non-coding RNAs (LncRNAs) were detected in this dataset. Many of these had altered expression patterns under cold stress compared to non-stress conditions. Our transcriptome results demonstrate that CRQ64 undergoes a wider stress response than CSQ5 under cold stress. Our results improved the annotation of the quinoa genome and provide new insight into the mechanisms of cold resistance in quinoa.

## 1. Introduction

The grain crop quinoa (*Chenopodium quinoa* Willd) is one of the oldest native crops. The seeds of quinoa have high levels of lipids, protein, and polyphenols, but have a low-glycemic index. Quinoa seeds are abundant in amino acids, including lysine, leucine, and isoleucine [1]. Additionally, quinoa has high resistance to abiotic stresses such as drought, cold, ultraviolet radiation, and salinity [2,3]. These advantages were recognized when the United Nations declared 2013 as the International Year of Quinoa [4].

Quinoa is typically grown in highland or cold climate areas. Because of the limitations of the climate in these regions, the sowing of quinoa is expedited to occur during the frost melts in flat areas. Subsequently, quinoa may suffer cold stress after sowing. Cold stress, an abiotic stress, has serious impacts on crop production. During cold stress, photosynthetic activity is inhibited, water uptake is reduced, and reactive oxygen species (ROS) are accumulated. Ultimately, the physiological homeostasis and metabolism of the plant are dysregulated [5,6,7]. Thus, cold tolerance is a necessary trait for many overwintering plants. Cold tolerance has been researched in many species, such as rice [8,9,10,11], *Sorghum bicolor* [12], and *Medicago falcate* [13]. Rice *COLD1* encodes a G-protein signaling protein that can interact with the G-protein α subunit to activate Ca^2+^ channels in low temperatures. Overexpression of *COLD1* significantly enhances chilling tolerance in rice [11]. Additionally, OsSAPK8 phosphorylates and activates a cyclic nucleotide-gated channel (OsCNGC9) to trigger Ca^2+^ influx. Thus, *OsCNGC9* positively regulates chilling tolerance by mediating cytoplasmic calcium influx [10]. The rice UDP-glycosyltransferase enzyme UGT90A1 is associated with low-temperature seedling survivability. Overexpression of *UGT90A1* helps to maintain membrane integrity during cold stress [8]. *Medicago falcata* has better tolerance to low temperature stress than alfalfa. Full-length transcriptome analysis of *M. falcate* roots identified 1538 transcription factor genes which had altered expression under cold stress. Furthermore, Cui et al. (2019) found that electrolyte leakage can be used as a marker of cell membrane damage caused by low-temperature stress [13].

Plant adaptation to stress conditions relies on the molecular networks involved in signal transduction and the expression of stress-related genes and metabolites [14]. Specifically, membrane stability is crucial for effective cold-stress resistance [15,16]. However, this protective mechanism is damaged by oxidative stress derived from ROS that are induced by cold temperatures. Transcription factors (TFs) regulate the expression of many cold stress responsive genes. *MYBs*, *WRKYs*, *AREB/ABF*, *NAC*, and *DREB1/CBF* (C-repeat binding factor) TFs regulate many stress responses, such as in response to salinity, temperature, and drought. In those *TFs*, *CBFs* are optimally representative genes whose functions have been verified in many plants [17,18,19,20]. Under cold stress, *CBFs* activate the expression of cold-responsive (COR) genes, including *COR15*, *COR78(RD29A)*, and *KIN1* [21,22,23]. *CBF* gene expression is itself controlled by many *TFs*, such as *ICE1*, *EIN3*, and *MYB15* [24,25]. The CBF signaling pathway promotes freezing tolerance [26]. Other TFs, such as apple *MYB308L*, also have key functions in cold stress. *MYB308L* complexed with *bHLH33* regulates cold tolerance and anthocyanin accumulation by activating the expression of *MdCBF2* and *MdDFR* in apple [27].

Though quinoa has cold resistance, the underlying molecular mechanisms are not clear. To obtain novel insight into the molecular mechanisms underlying the cold response of quinoa at the transcriptomic level, we performed transcriptome sequencing and gene expression profiling in normal-temperature-treated and cold-treated quinoa using the Oxford Nanopore Technologies (ONT) sequencing technique [28]. As a result, we identified numerous genes that may take part in the cold stress response. These genes are involved in plant development, primary and secondary metabolism, and hormone and signal transduction pathways. This study can provide insight into the cold stress response pathway in quinoa, and thus provide a basis for better general understanding of the plant cold stress response.

## 2. Results

### 2.1. Comparing Cold Sensitivity in CSQ5 and CRQ64 Quinoa Varieties

To compare cold sensitivity across different quinoa strains, we monitored the growing status of 50 quinoa varieties at different temperatures. Two typical representative strains were selected for further study: CSQ5 and CRQ64. CSQ5 and CRQ64 have similar growth performance at 22 °C (Figure 1A,B). To induce cold stress, CSQ5 and CRQ64 were cultivated in MS at 22 °C for 3 d and thereafter transferred to 4 °C. The control samples remained cultivated at 22 °C. After cultivating for 7 d at 4 °C, CSQ5 had closed cotyledons, short roots, and hypocotyls. CRQ64 had unfolded cotyledons and better growth performance than CSQ5 (Figure 1B,C). Additionally, under cold stress, CRQ64 had higher activity of superoxide dismutase (SOD), catalase (CAT), and peroxidase (POD), and lower malondialdehyde (MDA) content than CSQ5 (Figure 1D–H). Overall, these results showed that CRQ64 had better cold tolerance than CSQ5.

### 2.2. ONT Sequencing Enables the Sequencing of Full-Length Transcripts

Third-generation sequencing, such as the ONT method, has increased read lengths considerably compared to next-generation sequencing. ONT sequencing does not require fragmented RNA. Therefore, the sequencing read length is theoretically the full length of cDNA. We used ONT sequencing to analyze the transcriptomes of CSQ5 and CRQ64 after cold stress treatment for 24 h (CSQ5C and CRQ64C, respectively). CSQ5 and CRQ64 grown in normal conditions were used as the respective controls (CSQ5N and CRQ64N, respectively). We obtained approximately 9.1 million clean reads total in our 12 libraries, with an average read length of 862–1158 nucleotides for each library (Appendix A). The reads of this study have been deposited in the NCBI SRA database (PRJINA825645).

Over 99.4% of the ONT reads were successfully mapped to the quinoa reference transcripts identified from the reference genome (https://www.cbrc.kaust.edu.sa/chenopodiumdb/index.html, accessed on 19 April 2022) using Minimap2, indicating a relatively high integrity of the sequenced RNA. Full-length sequences accounted for on average 77.5% of the post-processed reads. When mapping the long reads to the existing annotated exons and untranslated regions (UTRs) in the quinoa reference transcripts, 92.3% (44,545 of 48,261) of the genes and 44.6% (44,583 of 99,974) of the isoforms were reproducibly recovered in four samples with three biological replicates (Appendix A). The differential mapping ratios of genes and isoforms suggest that some of the ONT reads may represent novel transcripts that are currently not annotated in the quinoa genome.

In total, 6432 novel genes were identified in our transcriptome. In these novel genes, 3716 novel genes were annotated. For example, the novel gene ONT.633 has three exons (122771431–122771684; 122771889–122772080; 122774100–122774629), a locus on chr0 corresponding to 122771431–122774629 in the quinoa genome. ONT.633 was annotated as a homolog of *Arabidopsis thaliana* exoribonuclease 2. Another novel gene, ONT.28735, is located at 74071107–74072332 on chr17 and was annotated as ribonuclease H protein. Many highly expressed novel genes were associated with stress-resistant functions. For example, ONT.27530 was identified as a homolog of AtGLK1b, which confers resistance to fungal and bacterial pathogens [29]. ONT.27860 was found to be a homolog of AtBIC1, which interacts with BRASSINAZOLE-RESISTANT1 (BZR1) and acts as a transcriptional coactivator in brassinosteroid signaling [30]. ONT.11394 was identified as a homolog of NRP1, a key gene promoting resistance to powdery mildew in cucumber [31]. In addition, ONT.28382 (GDSL esterase/lipase), ONT.27376 (carotenoid cleavage dioxygenase 4), and ONT.22508 (stress-associated protein 7) were also novel genes annotated as defense genes. These novel genes were detected herein for the first time in quinoa and had higher expression under cold stress in both CSQ5 and CRQ64. These genes may confer stress resistance in plants [32,33,34].

### 2.3. Novel Isoforms Identified by ONT Sequencing

We identified 55,389 novel isoforms in this transcriptome sequencing data (Appendix A). Most of these had at least 5 reads; only 2441 novel isoforms had less than 5 reads. All together, these novel isoforms represent a significant addition to the existing quinoa transcriptome. These novel isoforms include differential forms of alternative splicing and extensions or truncations at the 5′ or 3′ end.

Polyadenylated (Poly A) tails protect mRNA from exonuclease attack and have important functions in the termination of transcription, in the export of mRNA from the nucleus, and in the regulation of differential translation during development or under different environmental conditions. Alternative polyadenylation (APA) increases the diversity of the transcriptome and is involved in genome encoding and gene regulation. We analyzed the characteristics of APA sites in the different quinoa varieties. CSQ5 and CRQ64 have different numbers of APA sites. CSQ5 has fewer APA sites (39,930) than CRQ64 (42,157). In both CSQ5 and CRQ64, the number of APA sites was increased after cold treatment (Figure 2A). Thus, APA may be a form of splicing involved in the regulation of the cold stress response. Additionally, to search for a putative polyadenylation motif, we identified the top 10 APA motifs in our transcriptome data. These APA sites were enriched in the motif SRAG (Figure 2B), which accounts for 34.3% in the 10 main motifs.

### 2.4. Differentially Expressed Genes (DEGs)

In our transcriptomes, 44,608 genes, including novel genes, were identified. We analyzed the expression level of all genes in the four samples. Comparing the cold-stress response within each strain, we identified 3843 differentially expressed genes (DEGs) between CSQ5N and CSQ5C, and 5243 DEGs between CRQ64N and CRQ64C (Appendix A). In total, 2850 DEGs were shared between the two varieties, accounting for 74.16% of the CSQ5 DEGs and 54.36% of the CRQ64 DEGs, respectively (Figure 3A). These large numbers of DEGs indicate that the expression of the cold responsive genes is regulated at the transcript level. We also found there were more DEGs in CRQ64 than in CSQ5 under cold stress when compared to their respective controls, which may suggest a more complex response to cold stress in CRQ64, potentially connected to its cold resistance. In all DEGs, various stress-associated genes with a wide range of functions were detected, such as plant hormone signal transduction (e.g., abscisic acid receptor PYL6), TFs (e.g., ERF113), and antioxidase enzymes (e.g., peroxidase). Additionally, many flavonoid biosynthesis genes were differentially expressed in CRQ64C and CSQ5C, such as AUR62004178 (chalcone synthase 2), AUR62013677 (chalcone and stilbene synthase 6), and AUR62027465 (caffeoyl-CoA O-methyltransferase). These genes had higher expression in CRQ64 than in CSQ5 under cold stress (Appendix A). As ROS scavengers, flavonoids can inhibit ROS generation and reduce ROS levels [35,36]. Metabolism of flavonoids has been widely reported to be involved in the response mechanisms of plants to a wide range of stresses [37].

There were 2393 DEGs which were only present in CRQ64 (Figure 3A). These genes likely represent the determinants of the high cold resistance of CRQ64. Of these 2393 DEGs, the expression of 1339 genes were downregulated and 1054 were upregulated after cold stress. To validate some of these cold-stress-associated genes, 6 DEG unigenes were randomly selected for qRT-PCR verification. The expression patterns of these genes were consistent with our transcriptome data (Appendix A). 

In the 2393 CRQ64-specific DEGs, many genes which are important in regulating stimuli and stress responses, including CYP450s and heat shock protein (HSP), had differential expression in CRQ64 under cold stress. CYP450s can promote plant growth and protect plants from stresses through a variety of biosynthetic pathways, such as flavonoids, antioxidants, and phenylpropanoids. As molecular chaperones, HSPs protect plants against stress by reconstructing normal protein conformation [38]. Genes associated with stimuli and stress response, ROS balance, CYP450s, and HSP had similar expression patterns in CSQ5 and CRQ64 under cold stress, but had stronger changes in CRQ64 (Figure 3B).

We analyzed the enrichment of gene ontology (GO) terms (Appendix A) and KEGG pathways in the 2393 CRQ64-specific DEGs (Figure 3C). Enrichment analysis was performed on the biological process, molecular function, and cellular compartment levels (Appendix A). Generally, under cold conditions, plant vegetative growth and developmental progress are inhibited, and signal transduction and membrane activity are activated. In the 2393 CRQ64-specific DEGs, many genes were significantly enriched in starch and sucrose metabolism and phenylpropanoid biosynthesis (Figure 3C). In the KEGG class of these 2393 CRQ64-specific DEGs, many genes were enriched in endocytosis, plant–pathogen interaction, plant hormone signal transduction, and also starch and sucrose metabolism (Appendix A). Sucrose is an important component of carbon and nitrogen metabolism. Starch metabolism is known to be part of the cold stress response. Under cold stress, plants maintain an appropriate level of starch by regulating starch metabolism genes, such as *INDERMINATE DOMAIN 14* (*IDD14*) and *Qua-Quine Starch* (*QQS)* [39]. The accumulation of sucrose in plants often occurs concomitantly with environmental stresses including cold, drought, and salt [40,41]. Sucrose plays an important role in osmoregulation to maintain water potential in the plant [42,43]. In addition, sucrose can act as a crucial compound to regulate oxidative stress caused by environmental stress [44,45,46]. DEGs associated with starch and sucrose metabolism were partially upregulated in CRQ64 during cold stress, such as beta-fructofuranosidase (AUR62002193), hexokinase (AUR62010888), and polygalacturonase (AUR62002462), which may contribute to cold tolerance. Phenylpropanoid biosynthesis plays an important role in plant defense. Phenylpropanoid compounds may play important roles as signal molecules, both in plant development and plant defense. Specifically, phenylpropanoid compounds regulate oxidative metabolism and stress-induced cell death [47]. The glutathione metabolism pathway and flavone and flavonol biosynthesis pathway were also enriched in the CRQ64-specific DEGs. Glutathione participates in the maintenance of redox homeostasis [48]. Like flavonoids, glutathione metabolism is also a major mechanism for cellular protection against oxidative stress [49,50]. According to these results, we propose that starch and sucrose metabolism and ROS homeostasis are key mechanisms in CRQ64 for response to cold stress.

### 2.5. Analysis of Alternative Splicing Events and Corresponding Pathways in Response to Cold Stress

We analyzed five major types of alternative splicing events (exon skipping (ES), intron retention (IR), mutually exclusive exon (MXE), alternative 5′ splice sites (A5SS), and alternative 3′ splice sites (A3SS)) in our samples. IR was the most abundant alternative splicing event. In our transcriptome data, more than 30% of all alternative splicing events were IR, and about 25% were ES in CSQ5 and CRQ64 (Figure 4A). Under cold conditions, the distribution of the types of alternative splicing events was altered (Figure 4A). In both varieties, under cold conditions, the proportion of A3SS and IR events decreased while ES increased compared to their respective normal conditions. This indicated the potential impact of cold stress on promoting alternative splicing.

These newly identified stress-responsive splicing events were validated by RT-PCR. We randomly picked 4 known genes and 2 novel genes. As expected, the 6 examined genes showed consistent alternative splicing patterns with their profiles revealed by RNA-seq, confirming the accuracy of our bioinformatic analysis (Figure 4B). ONT.1201 (AUR62040272) and ONT.1509 (AUR62040994) had ES events in all samples. ONT.699 (AUR62042368), ONT.1250, and ONT.3787 had IR under cold stress. ONT.9047 (AUR62032383) had ES under cold stress. Some transcripts, such as ONT.1201.3 and ONT.3787.1, had low expression levels, and thus could not be detected. These results indicate the overall promotion of alternative splicing by cold stress.

There were 366 genes with differential alternative splicing (DAS) between CSQ5N and CSQ5C, and 537 between CRQ64N and CRQ64C, respectively (Appendix A). In total, 225 genes had DAS events in both CSQ5 and CRQ64 when comparing between cold and normal treatment. Of these 255 shared genes, many were annotated as peroxidases, such as AUR62037683 (peroxidase 18), AUR62013354 (peroxidase 25), and AUR62043592 (L-ascorbate peroxidase 3). Thus, peroxidases appear to participate in the response to cold in quinoa. To explore how many genes with DAS events in CRQ64 and CSQ5 also changed their expression under cold stress, a comparison was made between DEGs and genes with DAS events (Figure 4C). Only 132 genes with DAS events were also differentially expressed under cold stress. Most genes with DAS events (546) were not differentially expressed. In the 132 genes which were both alternatively spliced and differentially expressed, only 26 genes were both differentially expressed and alternatively spliced in both CSQ5 and CRQ64. Twenty-three genes had both differential expression and DAS events in CRQ64 specifically (Figure 4C). These 23 CRQ64-specific genes were mainly enriched in molecular function GO terms associated with catalytic activity, binding, cellular process, and transducer activity. Additionally, these 23 genes were enriched for plant hormone signal transduction, peroxisome, and fatty acid metabolism KEGG pathways. Therefore, CRQ64 may respond to cold stress by regulating plant hormone signaling, redox equilibrium, and fatty acid metabolism in a manner dependent on both AS and expression changes.

A total of 312 genes underwent DAS events in CRQ64 specifically (Figure 4C). Some heat shock proteins (HSP), such as AUR62007509 (HSP20), AUR62027958 (HSP90), underwent AS events in CRQ64 specifically but did not have differential expression under cold stress. IR caused a frame shift in AUR62007509 (HSP20) and ES in AUR62027958 (HSP90) resulted in a truncated coding region. Similarly, AUR62020806, which was annotated as RAP2-7 which might control subregulons of the downstream cold-responsive genes [51], also underwent DAS in CRQ64 specifically, but without expression changes under cold stress. ES also caused a truncated coding region in AUR62020806. Thus, DAS might cause the function of these genes to change under different environmental conditions.

To investigate the influence of stress-induced alternative splicing on cellular processes, we analyzed the functional categories (GO terms) and KEGG pathways of all genes with DAS events. We found differences between the annotations of genes with DAS events in CSQ5 and in CRQ64 when comparing between respective cold and normal treatments (Figure 5). The genes with DAS events in CRQ64 were enriched in more GO terms than those in CRQ5. In CRQ64 but not in CRQ5, genes with DAS events were associated with biological adhesion, extracellular region, cell junction, symplast, electron carrier activity, and nucleic binding transcription factor activity. Thus, under cold conditions, CRQ64 had a broader physiological response than CSQ5. In CRQ64, more fatty acid degradation and peroxisome-related genes were enriched than in CSQ5 (Appendix A). For example, AUR62042919 (3-ketoacyl-CoA thiolase 2) and AUR62042919 (3-ketoacyl-CoA thiolase 5) were enriched in CRQ64 only. These genes undergo DAS events to regulate fatty acid β-oxidation and ROS generation under cold conditions.

### 2.6. Genome-Wide Analysis of Cold-Stress-Associated Transcription Factors

Using iTAK software, we found that a total of 5988 genes in our data encoded functional domains of TF families (Figure 6A, Appendix A). In total, 374 TFs had differential expression after cold stress in CRQ64, and only 265 in CSQ5. A total of 187 TFs had altered expression in both CSQ5 and CRQ64 (Figure 6B). Many TFs are known to be responsive to stress, such as those in the MYB, C2H2, and AP2 families. AUR62035635 and AUR62012272 encode TFs belonging to the MYB family and had higher expression in CSQ5 under cold stress but were unchanged in CRQ64. AUR62002136 encodes a homolog of MYB4 and had altered expression in CRQ64 in response to cold stress. Overexpression of a *OsMYB4* in *Arabidopsis* enhances the expression of COR genes, proline levels, and freezing tolerance [52]. AUR62038383 encodes a ZAT12, a C2H2 zinc finger TF homolog, and had altered expression in CRQ64 under cold stress. *ZAT12* regulates 24 downstream genes, of which nine are cold-induced and 15 are cold-repressed genes in *Arabidopsis* [53]. Other TFs, such as AUR62004246 and AUR62004564, encoding bHLH family TFs, also had altered expression in CRQ64 in response to cold stress. *CBFs* have been extensively studied as key TFs involved in cold stress response in many plants [26]. In our data, AUR62021242 and AUR62016036 were homologous genes of *Arabidopsis CBFs*. The expression of AUR62021242 and AUR62016036 also had significant upregulation under cold stress in both CSQ5 and CRQ64. Overall, these results suggest that cold stress induced gene expression of TFs in a gene-family-dependent manner.

### 2.7. Distinct Expression Patterns of LncRNAs under Cold Stress

Non-coding RNAs have been reported to be involved in stress responses in plants. We found 2956 long non-coding RNA (LncRNA) transcripts using the online software Calculator (CPC), Coding-Non-Coding Index (CNCI), Coding Potential Assessment Tool (CPAT), and pfam protein structure domain. In the 2956 LncRNA transcripts, there were four groups: lincRNAs, antisense lncRNA, intronic lncRNA, and sense lncRNA. In total, 87% of the LncRNAs were lincRNAs (2577/2956) (Figure 6C).

The 2956 LncRNA transcripts correspond to 1671 genes (Appendix A). A total of 313 LncRNAs had altered expression after cold stress in CRQ64, but only 222 in CSQ5. Only 177 LncRNAs had differential expression in both CSQ5 and CRQ64 after cold stress (Figure 6D). More LncRNA transcripts were downregulated (82% and 81% of DEGs from CSQ5 LncRNAs and CRQ64 LncRNAs, respectively) than upregulated (18% and 19% of DEGs from CSQ5 LncRNAs CRQ64 LncRNAs, respectively) under cold stress, suggesting that cold stress repressed LncRNA expression.

We analyzed the target genes of the 136 LncRNAs which had differential expression in CRQ64 (Appendix A). Many target genes were associated with the phosphatidylinositol (PtdIns) signaling system. PtdIns signaling is related to Ca^2+^ signaling and regulates gene expression and responses to environment change. Osmotic stress induces rapidly synthesis of phosphatidylinositol-3,5-bisphosphate (PtdIns(3,5)P2), which might be a second messenger to response the hyperosmotic stresses in yeasts [54]. Thus, it is possible that quinoa resists osmotic stress caused by cold stress through the PtdIns signaling system in CRQ64.

## 3. Discussion

Plants cope with stress by adjusting their gene expression and metabolism to regulate their growth [55]. Quinoa is considered an important crop because of its exceptional resistance to harsh environmental conditions, such as salt, drought, and cold. Though the quinoa genome was assembled in 2017 [56,57], there has been little research on the quinoa transcriptome. Schmöckel et al. (2017) used a multifaceted approach integrating RNA-seq analyses with comparative genomics and topology prediction to identify 219 candidate genes that were differentially expressed in response to salinity in quinoa [58]. Other RNA-seq experiments have been used to reveal interactions between environment and quinoa [59,60]. Until now, the effects of cold stress on quinoa gene expression had not been explored. Thus, the molecular basis of cold tolerance in quinoa was still unclear. In this study, we used ONT sequencing technology to characterize gene regulation in response to cold stress in two quinoa varieties. In addition to obtaining high-quality full-length transcript sequences, we were also able to characterize AS events, transcription factors, and LncRNAs in quinoa and how they are affected by cold stress. Our data showed about 14.3% of protein-coding genes in total changed their expression and 5% had AS events in response to cold stress. The transcriptome data revealed that more genes had differential expression and alternative splicing in response to cold stress in the cold-tolerant CRQ64 strain than in the cold-sensitive CRQ5 strain, suggesting that alternative splicing can be used to respond to cold stress.

AS in plants participates in many biological processes, such as development, environmental responses, and the circadian clock [61,62,63]. Many studies have shown that IR is the most prominent AS event in plants [64,65], in agreement with our results. IR (32.8%), A3SS (23.2%), and ES (30.9%) were the predominant AS events in quinoa. In both quinoa varieties, under cold stress, A3SS and IR events were reduced while ES events were induced. In the genes that underwent DAS, few genes (19.5%) altered their expression as well in response to cold stress. We propose that changes in gene expression and AS events are regulated separately in the cold stress response in quinoa.

Splicing factors and their regulators play important roles in AS. We detected 189 genes associated with RNA splicing or regulation in this transcriptome. In these genes, only 6 had DAS events and 39 had altered expression after cold stress. These genes were annotated with the KEGG pathway spliceosome. In *Arabidopsis*, the widely studied Ser-Arg-rich (SR) proteins have been found to significantly influence AS processes. Previous work has also shown that loss of function of SR genes in *Arabidopsis* alters the splicing patterns of their own pre-mRNAs and those of several other genes. In quinoa, we detected 21 SR genes. *SR45a* (AUR62013860) underwent AS and two isoforms were detected. One of these isoforms changed its expression after cold stress. *SR30* (AUR62005213) underwent AS and five isoforms were detected. In the three isoforms of SR30 which had high expression, two isoforms changed their expression after cold stress. How the splicing factor/regulator genes are related to splicing patterns during cold stress warrants further study.

In this study, a total of 2393 DEGs were identified as responsive to low-temperature stress in CRQ64 specifically. In these genes, KEGG pathways analysis revealed that the antioxidant pathways, such as glutathione metabolism and flavone biosynthesis, were enriched in DEGs and peroxisome was enriched in AS (Figure 3C and Appendix A). We also analyzed the KEGG class of 2393 CRQ64-specific DEGs, many of which were enriched in the cellular process endocytosis (Appendix A). Endocytosis plays a key role in alleviating ROS [66,67]. In the organismal systems category, the DEGs were most enriched in plant–pathogen interaction and circadian rhythm–plant. Plants may have a basic immunological response under cold stress [15]. In the environmental information processing systems category, the DEGs were enriched in the plant hormone signal transduction pathway (Appendix A). Plant hormones play an important role in stress response [68,69]. In the genetic information processing category, many DEGs were enriched in the ribosome pathway (Appendix A). In the metabolism category, most DEGs were involved in carbon metabolism, starch and sucrose metabolism, and phenylpropanoid biosynthesis, demonstrating that energy metabolism is important for the cold stress response in quinoa (Appendix A). This may be related to plant photosynthesis, respiration, and protein translation. Photosynthesis is a highly sensitive process that responds to low temperature stress. Yang et al. (2019) reported that photosystem II reaction center PsbP family protein has a positive impact on oxygen release under high pH stress [70]. However, how photosynthesis-related genes are affected by cold stress is still poorly understood. 

Twenty-three genes had both differential expression and DAS in CRQ64 specifically (Figure 4C). For example, cold stress inhibited expression of AUR62003179, the splicing factor U2AF large subunit B, and induced its AS in CRQ64 specifically. Cold stress also promoted the expression and DAS of AUR62008295 and AUR62034166 (bZIP TFs). Transcriptome analysis comparing chilling tolerance in *indica* and *japonica* rice showed that bZIP TFs are induced by cold stress as a way to manage ROS [71]. AUR62008295 encodes a response factor 6 (ARR6), which can modulate plant cell wall composition and promote disease resistance [72]. ARR6 also responds to oxidative stress [73]. CRQ64 may respond to cold stress in a manner dependent on both AS and expression changes.

In conclusion, our study, for the first time, analyzed the full-length transcriptome of two quinoa varieties under cold treatment. The analysis revealed a total of 2850 cold-responsive DEGs and 225 genes with DAS events when CRQ64 and CSQ5 were pooled. There were 2393 DEGs and 312 genes that underwent DAS events in CRQ64 specifically. These genes were enriched in pathways related to ROS balance, such as starch and sucrose metabolism, glutathione metabolism and peroxisome, suggesting that ROS balance plays an important role in cold tolerance of CRQ64. These results improve our knowledge of the cold-induced transcriptional changes in quinoa. Furthermore, our transcriptome data enriches the full-length transcriptome and genetic data of quinoa. This research will contribute to future breeding of quinoa or other crops with increased cold tolerance.

## 4. Materials and Methods

### 4.1. Plant Cultivation and Cold Treatment

Quinoa CRQ64 and CSQ5 strains were used in this study. Quinoa seeds were sterilized using sodium hypochlorite solution (2.5%) and washed 5 times with sterile water. Sterilized seeds were planted on MS plates and subsequently put in the greenhouse at 22 °C with a photoperiod of 16 h light and 8 h dark. Three-day-old plants were transferred to a chamber for cold treatment at 4 °C with a 24 h dark cycle. The control seedlings were also cultured in the dark but at 22 °C. After 24 h of treatment, the treatment and control seedlings were collected and frozen in liquid nitrogen and stored at −80 °C for RNA extraction. Then, 1 µg total RNA was prepared for cDNA library preparation using the protocol provided by ONT.

### 4.2. Physiological Assays

The treatment and control seedlings were collected and weighed. The POD (Nanjing Jiancheng, #A084-3, Nanjing, China), SOD (Nanjing Jiancheng, #A001-1, Nanjing, China), CAT (Nanjing Jiancheng, #A007-1-1, Nanjing, China), Pro (Nanjing Jiancheng, #A107-1-1, Nanjing, China), and MDA (Nanjing Jiancheng, #A003-1, Nanjing, China) contents were measured. The soluble protein content was determined according to the Bradford method. All assays were repeated three times. 

### 4.3. RT-PCR and qPCR

A total of 1 µg total RNA was used for cDNA synthesis with PrimeScriptII 1st Strand cDNA Synthesis kit (TaKaRa, #6210A, Dalian, China) according to the manufacturer’s protocol. cDNAs were used for RT-PCR and qPCR. RT-PCR was performed to validate the alternatively spliced isoforms. qPCR was performed to validate the expression of DEGs using a LightCycler^®^ 96 Real-Time PCR System (Roche company, Basel, Switzerland). All primers used are listed in Appendix A.

### 4.4. Oxford Nanopore Technologies Long Read Processing

Raw reads were first filtered using a minimum average read quality score of 7 and a minimum read length of 500 bp. Ribosomal RNA were excluded after mapping to the rRNA database. Next, transcripts were determined by searching for primers at both ends of the reads. Clusters of FLNC transcripts were obtained after mapping to the reference genome (https://www.cbrc.kaust.edu.sa/chenopodiumdb/index.html, accessed on 18 April 2022) with mimimap2. Mapped reads were further collapsed by pinfish with min-coverage = 85% and min-identity = 90%. Duplicated transcripts were removed. Differences at the 5′ end were not considered when collapsing redundant transcripts.

### 4.5. Gene Functional Annotation and Functional Enrichment Analysis

Gene function was annotated based on the following databases: NR (NCBI non-redundant protein sequences); Pfam (protein family); KOG/COG/eggNOG (Clusters of Orthologous Groups of proteins); Swiss-Prot (a manually annotated and reviewed protein sequence database); KEGG (Kyoto Encyclopedia of Genes and Genomes); and GO (Gene Ontology).

GO enrichment analysis of the DEGs was implemented using the GOseq R package’s Wallenius non-central hyper-geometric distribution [74], which can adjust for gene length bias in DEGs.

KEGG is a database resource for understanding high-level functions and utilities of the biological system, such as the cell, the organism, and the ecosystem, from molecular-level information, especially large-scale molecular datasets generated by genome sequencing and other high-throughput experimental technologies (http://www.genome.jp/kegg/, accessed on 19 March 2021). We used KOBAS [75] software to test the statistical enrichment of the DEGs in KEGG pathways.

### 4.6. Quantification of Gene/Transcript Expression Levels

Full-length reads were mapped to the reference transcriptome. Reads with match quality above 5 were further used to quantify expression levels. Expression levels were estimated as reads per gene/transcript per 10,000 reads mapped. Differential expression analysis of two conditions was performed using the DESeq R package (2.0). DESeq provides statistical routines for determining differential expression in digital gene expression data using a model based on the negative binomial distribution. The resulting *p*-values were adjusted using the Benjamini and Hochberg’s approach for controlling the false discovery rate. Genes with an FDR < 0.01 and fold change ≥ 2 found by DESeq were assigned as differentially expressed.

### 4.7. Structure and Transcription Factors Analysis

Transcripts were validated against known reference transcript annotations with gffcompare. AS events including IR, ES, A3SS, A5SS, and MXE were identified by the AS talavista tool. APA analysis was conducted with TAPIS [76]. Plant transcription factors were identified with iTAK (http://bioinfo.bti.cornell.edu/tool/itak, accessed on 6 October 2021).

### 4.8. lncRNA Analysis

Transcripts with lengths of more than 200 bp and that had more than two exons were selected as lncRNA candidates and were further screened using CPC, CNCI, CPAT, and pfam protein structure domain analysis.

PfamScan: http://pfam.xfam.org/ (accessed on 6 October 2021)CPC: http://cpc.cbi.pku.edu.cn/ (accessed on 6 October 2021)CNCI: https://github.com/www-bioinfo-org/CNCI (accessed on 6 October 2021)CPAT: http://lilab.research.bcm.edu/ (accessed on 6 October 2021)

## Figures and Tables

**Figure 1 ijms-23-05724-f001:**
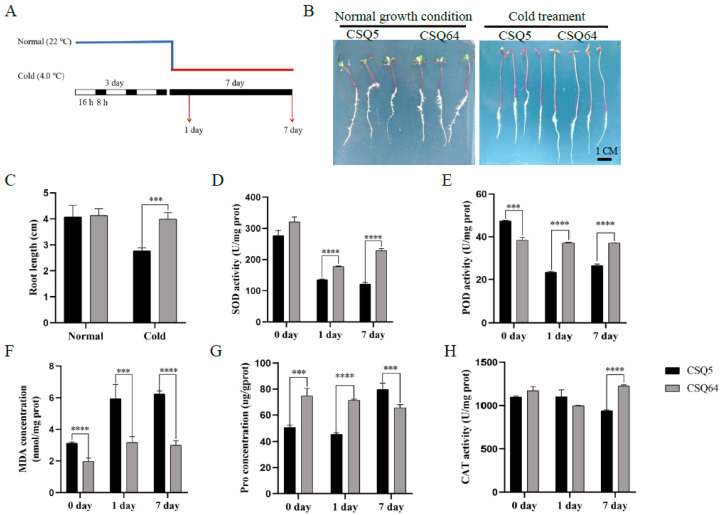
Cold sensitivity of CSQ5 and CRQ64. (**A**) Diagram of quinoa treatment. (**B**) Growing status of CSQ5 and CRQ6 in 22 °C for 3 d (left) and after transfer to 4 °C for 7 d in the dark (right). Scale bar = 1 cm. (**C**) Root length of CSQ5 and CRQ6 grown in 22 °C for 3 d and then grown in 22 °C (Normal) or 4 °C (Cold) for 7 d in the dark. (**D**–**H**) Physiological behaviors of CSQ5 and CRQ64 in cold stress. CSQ5 and CRQ6 were cultivated in MS at 22 °C for 3d and then transferred to 4 °C for 1d and 7d and tested for the various enzymatic activities and metabolite concentrations. ***, indicate significant differences of *t*-test (*p* ≤ 0.001) ****, indicate significant differences of *t*-test (*p* ≤ 0.0001).

**Figure 2 ijms-23-05724-f002:**
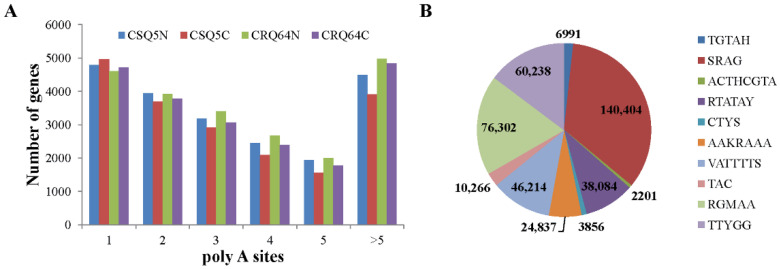
Distribution of poly A sites in CSQ5 and CRQ64. (**A**) Distribution of the numbers of poly A sites in genes found in the different strains and conditions; (**B**) motifs in the top 10 alternatively polyadenylated sites.

**Figure 3 ijms-23-05724-f003:**
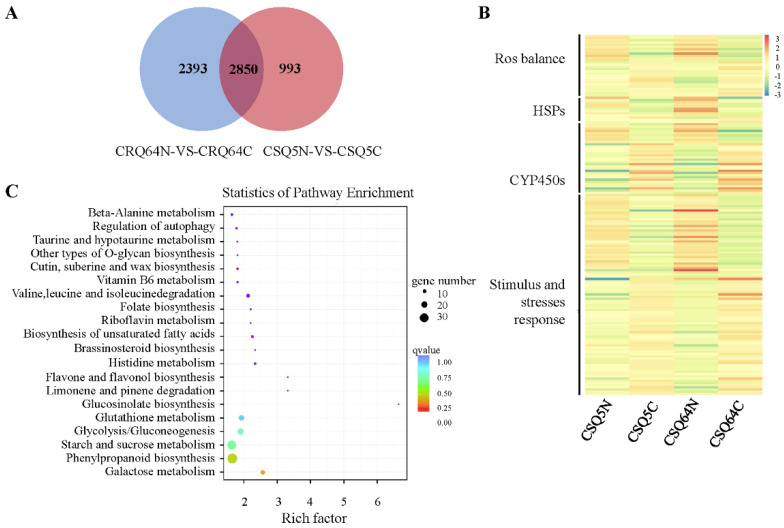
Analysis of DEGs in CSQ5 and CRQ64. (**A**) Venn diagram of DEGs. (**B**) Heatmap depicting expression pattern of genes associated with stimulus and stress response genes, ROS balance, CYP450s, and heat shock proteins (HSPs). The heatmap shows log2(FPKM) values of each protein. (**C**) KEGG enrichment of DEGs found in CRQ64 specifically.

**Figure 4 ijms-23-05724-f004:**
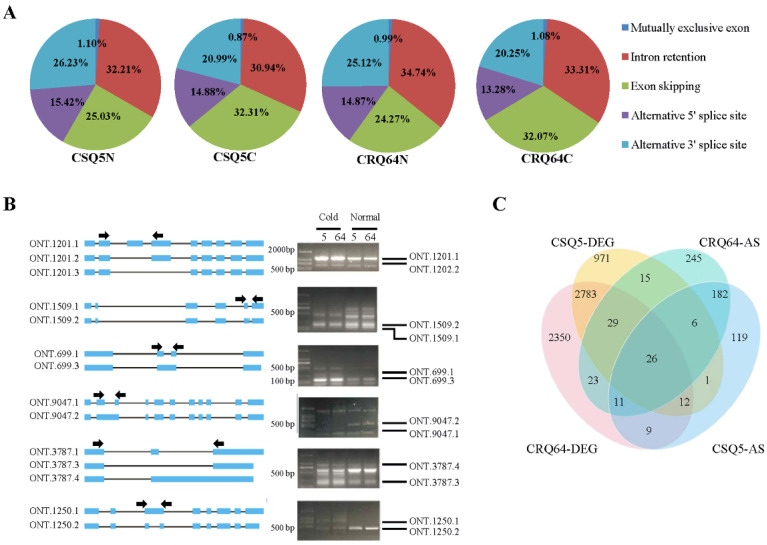
Distribution of alternative splicing in CSQ5 and CRQ64. (**A**) Distribution of types of alternative splicing in the different conditions. (**B**) RT-PCR verification of the AS events in CSQ5 and CRQ64. We selected four known genes AUR62040272 (ONT.1201), AUR62040994 (ONT.1509), AUR62042368 (ONT.699), and AUR62032383 (ONT.9047), and two novel genes, ONT.3783 and ONT.1250. Arrows indicate the positions of primers on selected genes. (**C**) Venn diagram of DEGs and genes with differential alternative splicing (DAS).

**Figure 5 ijms-23-05724-f005:**
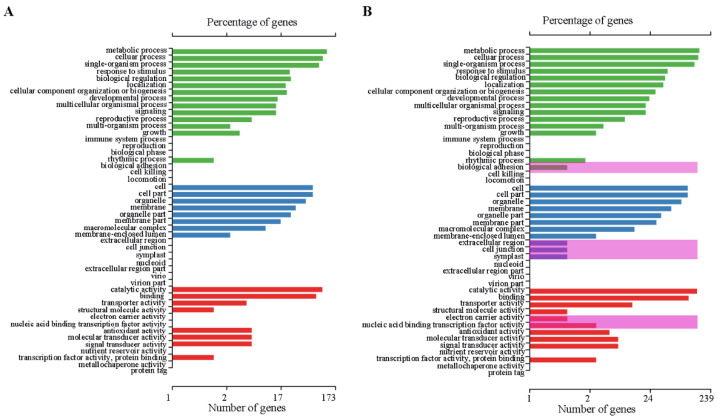
GO enrichment of the genes with DAS events in CSQ5 and CRQ64. (**A**) GO enrichment of the genes with DAS events between CSQ5N and CSQ5C. (**B**) GO enrichment of the genes with DAS between CSQ64N and CSQ64C.

**Figure 6 ijms-23-05724-f006:**
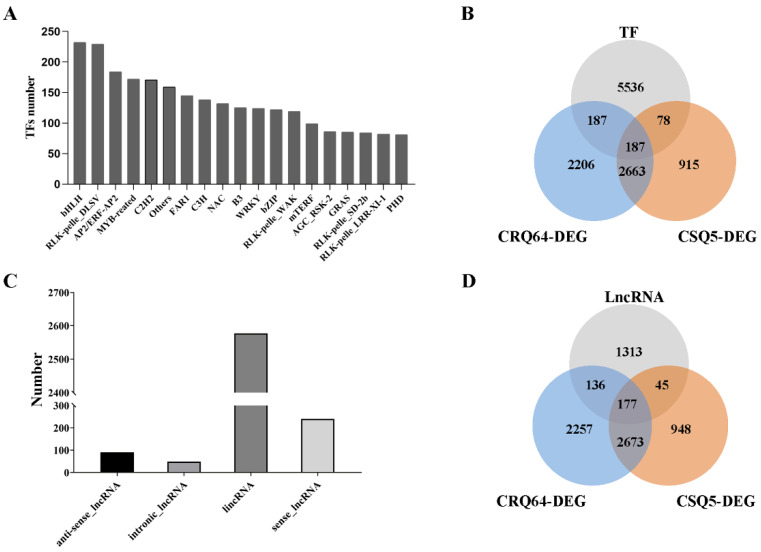
TFs and LncRNAs analysis in CSQ5 and CRQ64. (**A**) The gene numbers of TF families in this transcriptome. (**B**) Venn diagram of TFs and DEGs in CSQ5 and CRQ64 under cold stress. (**C**) Number of LncRNAs transcripts in this transcriptome. (**D**) Venn diagram of LncRNAs transcripts and DEGs in CSQ5 and CRQ64 under cold stress.

## Data Availability

The RNA-seq data have been submitted to the National Center for Biotechnology Information Sequence Read Archive (http://www.ncbi.nlm.nih.gov/sra, accessed on 19 April 2022) under BioProject accession numbers PRJINA825645.

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
