# Peer review of "Full-Length Transcriptome Sequencing Reveals the Impact of Cold Stress on Alternative Splicing in Quinoa"

_ijms, 2022, doi:10.3390/ijms23105724_

Round 1

Reviewer 1 Report

Comments of Reviewer #

Manuscript ID: ijms-1713654

“Full-length transcriptome sequencing reveals the impact of cold stress on alternative splicing in quinoa” by Zheng et al.

General comments

In this article, the authors analyzed the differences in whole transcriptome profiles of two quinoa varieties (resistant and sensitive to low temperature, respectively) under cold stress conditions. They found that the resistant variety (CRQ64) more differentially responds to gene expression and alternative splicing (AS) than does the sensitive one (CSQ5). Especially, starch and sucrose metabolism and ROS homeostasis were found to have a key role in response to cold in CRQ64 in DEG analyses; GO term analysis revealed that peroxidase functions are enriched in AS event of cold response in CRQ64. They also analyzed the roles of long non-coding RNAs and transcription factors in responses under cold stress. Those comprehensive and panoramic views of transcriptomic analysis will provide a novel view to understand how plants (quinoa) survive under unfavorable conditions.

   This reviewer thinks that this manuscript is acceptable to publish on IJMS. However, this reviewer has some questions and minor comments that are expected to respond from the authors.

Questions

AS is a well-known event under stress conditions in Drosophila as well as Arabidopsis. Is AS simply controlling the amounts of functional protein (ie., truncated protein may be unfunctional, so that total amounts of functional protein are reduced) or does it provide another functional protein (ie., generation of alternative protein isoforms that have their function)? Do you have any idea how AS functionally works under cold stress in a plant (quinoa)?

Minor comments

1) Fig.4B is difficult to follow. Please replace a more complaisant one.

2) L.285-286. …in CSQ5 than in CRQ64. Is this “…in CSQ5 and in CRQ64” or “…in CSQ5 from in CRQ64”?

3) L.330. There is an extra space between “domain” and “.”.

4) Following references are preferable to cite in the discussion of the role of AS under stress conditions of plants.

Filichkin et al. (2015) Curr Opin Plant Biol. 24 :125

Martin et al. (2021) Genome Biol. 22:35

Liu et al. (2022) Front Plant Sci https://doi.org/10.3389/fpls.2022.832177

Author Response

Dear editors,

Thank you for giving us the opportunity to revise our manuscript entitled “Full-Length Transcriptome Sequencing Reveals the Impact of Cold Stress on Alternative Splicing in Quinoa”. I appreciate the constructive comments and suggestions from the reviewers. In this resubmission, we have addressed all comments and concerns point-by-point and revised our manuscript accordingly. In addition, Qun Shao, who had participated in revised this project, was added as a co-author in the revised manuscript.

I hope the you and the reviewers will find  the revised manuscript suitable for publication. I look forward to hearing from you soon.

Yours Sincerely,

College of Life Sciences

Shandong Normal University

Jinan, 250014, PR China

Point-by-point Responses

Reviewers' comments:

Reviewer #1: General comments

I   This reviewer thinks that this manuscript is acceptable to publish on IJMS. However, this reviewer has some questions and minor comments that are expected to respond from the authors.

Questions

AS is a well-known event under stress conditions in Drosophila as well as Arabidopsis. Is AS simply controlling the amounts of functional protein (ie., truncated protein may be unfunctional, so that total amounts of functional protein are reduced) or does it provide another functional protein (ie., generation of alternative protein isoforms that have their function)? Do you have any idea how AS functionally works under cold stress in a plant (quinoa)?

Response: AS events often introduce premature stop codons in plants, generating truncated isoforms. In some cases, truncated isoforms can still interact with the same target to act as negative regulators. In quinoa, as in other plants, many AS events are induced under environmental stress. Under cold stress in quinoa, many genes with AS events produce new splicing variants. The functions of these genes that undergo AS, potentially regulated by the ratio of the different isoforms, could change under cold stress. We speculate that the truncated transcripts can competitively interact with the same targets. Thus, the ratios of the different splice variants are a critical mechanism to regulate gene function in response to cold stress. However, uncovering the specific mechanisms underlying how AS responses to cold stress in quinoa will require further studies.

Minor comments

1) Fig.4B is difficult to follow. Please replace a more complaisant one.

Response: We have revised Figure 4B. In the new figure, only ONT numbers were used.

2) L.285-286. …in CSQ5 than in CRQ64. Is this “…in CSQ5 and in CRQ64” or “…in CSQ5 from in CRQ64”?

Response: In this sentence, it should be “…in CSQ5 and in CRQ64”. We had changed it in our revised manuscript.

3) L.330. There is an extra space between “domain” and “.”.

Response: changed.

4) Following references are preferable to cite in the discussion of the role of AS under stress conditions of plants.

Filichkin et al. (2015) Curr Opin Plant Biol. 24 :125

Martin et al. (2021) Genome Biol. 22:35

Liu et al. (2022) Front Plant Sci https://doi.org/10.3389/fpls.2022.832177

Response: Thanks for your advice. These references were cited in the revised MS.

Reviewer 2 Report

The manuscript, “Full-Length Transcriptome Sequencing Reveals the Impact of Cold Stress on Alternative Splicing in Quinoa”, was studied by Zheng et al. The authors analyzed the full-length transcriptomes of the cold-resistant CRQ64 and cold-sensitive CSQ5 in Quinoa varieties. The results revealed that 55,389 new isoforms and 6,432 novel genes were identified via transcriptomic analysis. Interestingly, cold-treated CRQ64 harbored more DEGs such osmoregulation and ROS homeostasis as well as alternative splicing products such as peroxidase-related genes in comparison with those in CSQ5 under control condition. Although the mechanism of the alternative splicing patterns and the regulation of photosynthesis-related genes in response of plant to cold stress need to be further elucidated, the authors provided solid notions and sentences with the accordance to their results. I have several minor comments that should be addressed.

  1. In line 45, “interacts” should be corrected by “interact”.
  2. In line 54, please add year after Cui et al.
  3. In line 124, please confirm the word of “chr0”.
  4. In line 331, “87% of” can be rewritten with “Eighty seven percent of”.
  5. In line 336-337, please denote down- and up-regulated genes with percent as well.
  6. In line 344, “resist” should be corrected by “resists”.
  7. In line 351, please add year after Schmöckel et al.
  8. In line 416-425, please move the paragraph into the section of conclusion.
  9. In line 419, “There are” should be corrected by “There were”.
  10. In line 420, is it true that starch and sucrose metabolism are involved in the ROS balance?
  11. In line 421, “play” should be corrected by “plays”.
  12. In line 422, “These result improves” should be corrected by “These results improve”.
  13. I would like to recommend the authors to re-read the manuscript, thereby rectifying the grammatical and typographical errors.

Author Response

Reviewer #2: The manuscript, “Full-Length Transcriptome Sequencing Reveals the Impact of Cold Stress on Alternative Splicing in Quinoa”, was studied by Zheng et al. The authors analyzed the full-length transcriptomes of the cold-resistant CRQ64 and cold-sensitive CSQ5 in Quinoa varieties. The results revealed that 55,389 new isoforms and 6,432 novel genes were identified via transcriptomic analysis. Interestingly, cold-treated CRQ64 harbored more DEGs such osmoregulation and ROS homeostasis as well as alternative splicing products such as peroxidase-related genes in comparison with those in CSQ5 under control condition. Although the mechanism of the alternative splicing patterns and the regulation of photosynthesis-related genes in response of plant to cold stress need to be further elucidated, the authors provided solid notions and sentences with the accordance to their results. I have several minor comments that should be addressed.

Minor comments

    1.In line 45, “interacts” should be corrected by “interact”.

Response: We have changed it in the revised manuscript.

2.In line 54, please add year after Cui et al.

Response: changed.

3.In line 124, please confirm the word of “chr0”.

Response: “chr0” is correct. The chromosome information was obtained from quinoa information resource https://www.cbrc.kaust.edu.sa/chenopodiumdb/index.html. Chr0 indicates genome sequence that cannot assign to any know chromosome in quinoa.

4.In line 331, “87% of” can be rewritten with “Eighty seven percent of”.

Response: We did not change it since 87% of” and “Eighty seven percent of” have the same meaning.

5.In line 336-337, please denote down- and up-regulated genes with percent as well.

Response: The percentage of down- and up-regulated genes were added.

6.In line 344, “resist” should be corrected by “resists”.

Response: We have changed it in revised manuscript.

7.In line 351, please add year after Schmöckel et al.

Response: corrected.

8.In line 416-425, please move the paragraph into the section of conclusion.

Response: We want to keep the conclusion section as concise as possible, therefore, we didn’t change it.

  1. In line 419, “There are” should be corrected by “There were”.

Response: corrected.

10.In line 420, is it true that starch and sucrose metabolism are involved in the ROS balance?

Response: About the starch and sucrose metabolism and ROS balance, we have cited articles for reference. Many researchers found sugars are involved in oxidative stress and had positive functions in ROS stress.

Ivan, C.; Cécile, S.; Gwenola, G.; Abdelhak, E.A. Involvement of soluble sugars in reactive oxygen species balance and responses to oxidative stress in plants. J. Exp. Bot. 2006, 449-459.

Chiou, D.; Bush, D. Sucrose is a signal molecule in assimilate partitioning. Proc. Natl. Aacd. Sci. U.S.A. 1998, 95, 4784-4788.

Valluru, R.; Van, D.E.V. Sucrose, sucrosyl oligosaccharides, and oxidative stress: scavenging and salvaging? J. Exp. Bot. 2009, 60, 9-18.

11.In line 421, “play” should be corrected by “plays”.

Response: We have changed it in revised manuscript.

12.In line 422, “These result improves” should be corrected by “These results improve”.

Response: changed

    13.I would like to recommend the authors to re-read the manuscript, thereby rectifying the grammatical and typographical errors.

Response: We have improved the quality of English and we had rectified the grammatical and typographical errors.